# copan:LPJmL: A new hybrid modelling framework for dynamic land use and agricultural management

Jannes Breier<sup>1,2,3,\*</sup>, Luana Schwarz<sup>1,3,4</sup>, Hannah Prawitz<sup>1,3,5</sup>, Werner von Bloh<sup>1</sup>, Christoph Müller<sup>1</sup>, Stephen Björn Wirth<sup>1</sup>, Max Bechthold<sup>1,3,5</sup>, Dieter Gerten<sup>1,2</sup>, and Jonathan F. Donges<sup>1,3,6</sup>

**Correspondence:** Jannes Breier (jannes.breier@pik-potsdam.de)

- Abstract. Dynamic Global Vegetation Models (DGVMs) are established in environmental and agricultural sciences for many
- purposes, e.g., modelling plant growth and productivity, water and carbon cycles, and biosphere-climate interactions. Never-
- theless, DGVMs are still rather limited in terms of simulating mutual interactions between biospheric and human processes.
- While DGVMs such as the Lund Potsdam Jena managed Land (LPJmL) model have been successfully connected to Integrated
- Assessment Models (IAMs), the model couplings often remain loose and static over the simulation period. The copan:LPJmL
- modelling framework is an extension of the copan:CORE framework for integrated and dynamic human-Earth system mod-
- elling, and addresses this issue by integrating LPJmL via a new interface, consisting of an LPJmL coupling library and a Python
- library pycoupler, which together enable LPJmL inputs and outputs to be coupled in copan:LPJmL during the simulation pe-
- riod. It uses the copan:CORE entities and integrates the coupled data into the World (simulation space as a whole) and the
- (grid) Cell entity, allowing other entities such as Individuals, e.g., for agent-based modelling (ABM), to access them. Besides
- ABM, this framework allows for a broad range of modelling approaches to be represented with copan:LPJmL, of which we
- introduce three examples: (1) The model of Integrated Social-Ecological Resilient Land Systems (InSEEDS), which uses a
- classical ABM approach to model management decisions by farmers, (2) an adaption of an established crop calendar model
- (Crop Calendar), and (3) a novel Large Language Model (LLM)-driven ABM approach (LLM Fertilization).

<sup>&</sup>lt;sup>1</sup>Potsdam Institute for Climate Impact Research (PIK), Member of the Leibniz Association, P.O. Box 60 12 03, 14412 Potsdam, Germany

<sup>&</sup>lt;sup>2</sup>Humboldt Universität zu Berlin, Department of Geography, Unter den Linden 6, 10099 Berlin, Germany

<sup>&</sup>lt;sup>3</sup>Integrative Earth System Science, Max Planck Institute of Geoanthropology, Kahlaische Str. 10, 07745 Jena, Germany

<sup>&</sup>lt;sup>4</sup>University of Potsdam, Institute of Environmental Science and Geography, Karl-Liebknecht-Str. 24-25, 14476 Potsdam, Germany

<sup>&</sup>lt;sup>5</sup>University of Potsdam, Institute of Physics and Astronomy, Karl-Liebknecht-Str. 24-25, 14476 Potsdam, Germany

<sup>&</sup>lt;sup>6</sup>Stockholm Resilience Centre, Stockholm University, Kräftriket 2B, 106 91 Stockholm, Sweden

#### 1 Introduction

15

The Anthropocene is a new epoch in the evolution of the Earth system characterized by increasingly strong and entangled 16 17 coevolutionary dynamics of biogeophysical Earth system processes with human societies (Steffen et al., 2011; Schellnhuber, 1999; Crutzen, 2002). The recognition of biogeophysical and social processes as intertwined has emerged from historical 18 developments in the field of Earth System Science. While James Lovelock already published the Gaia hypothesis in the early 19 1970s (Lovelock and Margulis, 1974), the broader scientific uptake of the term "Earth System Science" took place in the 1980s 20 21 and 1990s (Lenton, 2016; Steffen et al., 2020). Central publications in this decade, such as the Bretherton Diagram (Committee, 1986) and the Brundtland report (Holdgate, 1987), acknowledge that human societies are tightly connected to Earth system 22 dynamics. Followed by the realisation that (some) human societies now act as the major driving force of change on our planet 23 (Steffen et al., 2011), while at the same time being shaped and impacted by the ecological conditions they are embedded in 24 25 (Rockström et al., 2009), diverse novel conceptualisations of this intertwinedness, like the "technosphere" (Rosol et al., 2017) 26 have emerged. 27 The consideration of the role of humans in the Earth System has thus progressed towards a coevolutionary, bidirectional approach, now evident in different strands of simulation modelling that can be summarised under the term World-Earth (System) 28 modelling (WEM) or integrated human-Earth system modelling, with the entirety of human civilizations being referred to as 29 30 the World (Beckage et al., 2020; Donges et al., 2021). There is growing research calling for and adopting a social-ecological, integrated perspective of humans embedded in the Earth System (e.g., (Donges et al., 2017; Schill et al., 2019; Beckage et al., 31 32 2020; Farahbakhsh et al., 2022; Beckage et al., 2022; Moore et al., 2022; Gerten et al., 2018). Different modelling communities approach this integration in distinct ways, and focus on different aspects of social-ecological dynamics. One central challenge 33 34 of these efforts is moving beyond simple proof-of-concept models towards more complex, integrated models (Beckage et al., 2020). 35 36 Land systems are a key example of the inherently coevolutionary nature of social-ecological interactions in World-Earth 37 Systems (Meyfroidt et al., 2022). Agricultural production is the single largest driver of transgressions of multiple planetary boundaries: land-system change, freshwater use, biogeochemical flows and biosphere integrity (Campbell et al., 2017), also 38 contributing to the transgression of other planetary boundaries, such as climate change. In socio-ecological terms, global food 39 40 demand drives responses in land use and production, including cropland and pasture expansion, and intensification through management practices such as irrigation and fertilization, and the reorganization of supply chains through trade. On the pro-41 duction side, farmers' management decisions, such as the use of crop rotations, cover cropping, reduced tillage, and integrated 42 pest management can significantly influence the adverse environmental impacts that manifest themselves in the transgression 43 of planetary boundaries (Gerten et al., 2020). Improved knowledge about the dynamic and coevolutionary development of agri-44 45 cultural systems at scales from local to global is therefore imperative to foster our understanding of fundamental Anthropocene dynamics. 46 47 Land use systems alone have been a research subject in Earth system science for years, leading to their incorporation into 48 DGVMs (such as LPJmL), crop models (e.g., DSSAT) as well as Earth System Models (ESMs, (e.g. MPI-ESM)) (Foley

49 50

59

61

63

70

73

76

78

et al., 2005; Bondeau et al., 2007; Jones et al., 2003; Reick et al., 2013). Specific (such as tillage) or bundled management practices (like conservation agriculture) have been represented in detail and studied in local and global applications of such models (Herzfeld et al., 2021; Ngwira et al., 2014). Additionally, more holistic applications have been carried out, e.g., to show the potential of maintaining multiple planetary boundaries while feeding 10 billion people (Springmann et al., 2018; Gerten et al., 2020). However, the most important factor in implementing such measures or achieving such goals has been neglected: Humans are often merely represented as biophysical 'disturbance' factors. This way, their decision-making capacities and social-psychological and socio-cultural complexities are therefore underrepresented, with many assumptions and dynamics, for example on land management and land use, remaining rather static (Schellnhuber, 1999; Rounsevell et al., 2014; Beckage et al., 2022). IAMs make it possible to simulate and optimize certain land use dimensions under a set of "human boundary conditions", such as Shared Socioeconomic Pathways (SSP) scenarios (Dietrich et al., 2019). Through the application of macroeconomic and energy-economic optimization approaches, many IAMs reduce the complexity of human behaviour to predefined scenarios or boundary conditions, thereby excluding key dimensions of human decision-making uncertainty (Beck and Krueger, 2016; Asefi-Najafabady et al., 2021; Koasidis et al., 2023). While the SSP1 and SSP2 scenarios presented in the Sixth IPCC Assessment Report are considered economically feasible within IAMs, their assumptions regarding socialecological feasibility—such as behavioural adaptation and governance dynamics—remain subject to significant uncertainty (int, 2023; Schleussner et al., 2024; Krawczyk and Braun, 2025). Furthermore, these scenarios lack the fully coupled feedbacks of societies with the Earth system, i.e. the coevolution inherent in the underlying dynamics that is currently not representable endogenously with the majority of existing modelling approaches (Schlüter et al., 2012; Calvin and Bond-Lamberty, 2018). To address this gap, Donges et al. (2020) introduced the copan:CORE modelling framework, which supports the development of World-Earth models (WEMs). WEMs are characterised by the explicit and bidirectional coupling of social and Earth system processes, enabling the coevolution of human and natural systems to be represented within a single modelling framework. They aim to go beyond traditional IAMs by incorporating human agency, social heterogeneity, and feedbacks between human decisions and biophysical dynamics (Mathias et al., 2020). A model built in the copan:CORE modelling framework consists of entities, such as a simulation cell or individuals, that interact with each other via various processes (Donges et al., 2020). The latter are categorized by three overlapping process taxa, representing biogeophysical and biogeochemical (ENV, e.g. biophysical conditions, crop growth), socio-metabolic (MET, e.g. crop harvest, fertilization) and socio-cultural processes (CUL, e.g., governance, social learning, social norms dynamics, or individual cognitive-behavioural processes like attitude formation) (Donges et al., 2021). We here have advanced this framework by integrating LPJmL as the ENV taxon (Schaphoff et al., 2018), through which we can represent diverse natural as well as managed land systems, such as forests, grassland, and cropland (Sakschewski et al., 2016; Wirth et al., 2024b; Braakhekke et al., 2019; Rolinski et al., 2018; Porwollik et al., 2021; Jägermeyr et al., 2015; Minoli et al., 2022a) and link them bidirectionally to social processes that can be of economic nature but also go beyond that, for example including behavioural change, collective decision making or political processes. Based on this, MET and CUL-based components of different modelling approaches can be realized by applying copan:LPJmL to represent various dimensions of social-ecological systems. By building on the diverse entities and process/model templates of

**Figure 1.** Basic scheme of copan:LPJmL featuring the global grid, the process taxonomy as well as main entities with the three application examples: InSEEDS, Crop Calendar, and LLM Fertilization.

copan:CORE, complex and detailed systems can be integrated to map underlying dynamics such as human, social, and societal processes using ABMs, dynamical system, or detailed rule-based approaches.

In this paper, we introduce the copan:LPJmL modelling framework as a flexible and extensible platform for the development of WEMs, including a process-based DGVM Earth in a land system context. To demonstrate its applicability and versatility, we present three example applications: (1) nSEEDS, a global agent-based framework for simulating the adoption of regenerative agricultural practices by farmer agents (Schwarz et al.); (2) Crop Calendar, an integrated (in runtime calculation) update of the rule-based model published by Minoli et al. (2019, 2022a); and (3) LLM fertilization, representing the application of crop fertilizer by LLM farmer agents. These examples illustrate how copan:LPJmL enables the integration of diverse socio-cultural and socio-metabolic processes with detailed Earth system dynamics.

### 2 Framework description

copan:LPJmL is a newly developed, enhanced WEM framework that integrates the DGVM LPJmL and the existing open modelling framework copan:CORE, thereby creating a modelling environment in which biophysical Earth-system and social world processes can be represented and bidirectionally linked to enable fully integrated and global gridded social-ecological modelling. It is designed to bring together the details and complexity of a process-based land biosphere model with the flexibility of an open modelling framework, thereby enabling the build-up of new types of integrated models. It allows selection from vari-

104

123

ous modelling approaches such as agent-based, rule-based, dynamical system, or data-driven (statistical). copan:CORE forms the core framework in which the modular structure and building blocks, such as basic entities, are defined that can interact with each other. In copan:LPJmL, LPJmL is currently integrated via annual coupling to represent processes of the ENV taxon, with future development aiming to implement more frequent coupling intervals, ranging from monthly to daily. This is facilitated by LPJmL itself, which has been extended with a coupler library that is available in LPJmL from version 5.6 onwards, and applied in copan:LPJmL via the Python interface pycoupler (Breier and von Bloh, 2025).

#### 2.1 The CORE framework

copan:CORE, introduced by Donges et al. (2020), has established a novel systematic approach for building WEMs in a Python-105 based modelling environment with its corresponding Python library pycopancore. All copan:CORE model elements are rep-106 resented as modular, object-oriented entities. These include agents (such as households, firms, or governments), social institu-107 108 tions, components of the Earth system or even more abstract entities such as a rulebook (basis for Crop Calendar). Additional entities can be flexibly introduced using the framework. In copan:CORE, each of these entities is involved in model processes 109 that can be categorized by applying the aforementioned taxonomy system described by Donges et al. (2021) (ENV, MET 110 and CUL). Another essential feature of copan:CORE is its support for heterogeneous agent populations and scalable network 111 structures that define the relationships and interactions among agents, referred to as Individuals, and between agents and 112 their linked entities. Besides Individuals these available entities in copan: CORE are the simulation space, the World, 113 the underlying elemental spatial units, the Cells, the Social Systems such as countries or cities (Donges et al., 2020). 114 Recently, Bechthold et al. (2025) added an additional Group entity as a social structure in which Individuals can organize 115 themselves. For copan:LPJmL, this variety is especially valuable, as the underlying hierarchies and networks provide a basic, 116 real-life reflecting structure that aligns well with the simulation space (world) of LPJmL, in which Cells also reflect the 117 elementary spatial unit of simulation. This way, the framework can represent the LPJmL organizational structure within the 118 copan:CORE World. The library, pycopancore, also included model components, models and studies outside of the CORE 119 120 definition (tillkolster et al., 2020). Within the process of building new software around copan:CORE, such as copan:LPJmL, this structure was reorganized to distinguish the CORE from other software and components related to copan:CORE, leaving 121 only the components used by Donges et al. (2020) as an exemplary modelling approach (Breier et al., 2025b). 122

#### 2.2 LPJmL as a model component

The DGVM LPJmL simulates the carbon, nitrogen, water and energy cycles of the terrestrial biosphere in coupling with the growth and productivity of natural and agricultural ecosystems, forced by climate, crop distribution, and other globally gridded input data (Schaphoff et al., 2018; von Bloh et al., 2018; Wirth et al., 2024a). It can represent various agricultural management practices that require additional information on the spatial and temporal distribution of management systems (Lutz et al., 2019; Porwollik et al., 2021; Jägermeyr et al., 2015; Herzfeld et al., 2021; Minoli et al., 2022a; Jägermeyr et al., 2016; Heinke et al., 2023). The smallest entity representing these processes is a grid cell, by default with a spatial extent of  $0.5 \times 0.5$  degrees, such that 67,420 cells represent the land surface globally. Plants are modelled according to the concept of Plant Functional

Figure 2. Stylized class diagram of the integrated copan:LPJmL WEM framework. copan:LPJmL (white, dotted box) consists of various software components from which different functions and/or classes are used: LPJmL (green box) is linked via pycoupler (yellow), which is responsible for configuration, simulation, and exchange of data in the xarray format (gray box) with copan:LPJmL. World and Cell, as well as model components, are child classes of the corresponding copan:CORE entities and components.

- Types (PFT) (Smith et al., 1993). The same principle is applied to Crop Functional Types (CFT) used for modelling agriculture on prescribed shares of each cell, called stands (Bondeau et al., 2007). copan:LPJmL introduces LPJmL as a new model component that covers the ENV taxon. It represents the LPJmL model as a World entity  $W_{LPJmL}$  (LPJmL simulation space,
- i.e. global), that integrates the coupled LPJmL model

$$\mathcal{W}_{LPJmL} = (LPJmL, \{C_i \mid i = 1, 2, \dots, n\})$$

$$\tag{1}$$

and its corresponding Cell entities  $C_{\text{LPJmL}}$  as shown in Fig. 2 following the concept of Donges et al. (2020).  $C_{\text{LPJmL}}$  includes the input and output of LPJmL for the cell as well as the grid information (longitude, latitude) and a network of its neighbouring

cells  $\mathcal{N}(C_i)$ :

151

$$C_i = (\text{input}, \text{output}, \text{grid}, \mathcal{N}(C_i)),$$
 (2)

where  $\mathcal{N}(C_i)$  is defined as

$$\mathcal{N}(C_i) = \{ C_i \mid d(C_i, C_j) \le r, j \ne i \text{ and } C_i \in \mathcal{W}_{LPJmL} \}$$
(3)

and is calculated based on the grid information of each cell, with  $d(C_i, C_j)$  denoting the Euclidean distance between cell centroids, and  $r = \sqrt{2} \cdot 0.5^{\circ}$  corresponding to the maximum distance between adjacent cells in the  $0.5^{\circ}$  grid (including diagonal 140 neighbours). In the copan:LPJmL implementation, both the World entity and its corresponding Cell entities are realized 141 using efficient data structures provided by Python libraries xarray and numpy, which enable both array-based and object-142 oriented access (Hoyer and Hamman, 2017; Harris et al., 2020). The attributes of an instantiated World (world, lower case), 143 144 such as LPJmL input, output, and grid information, are stored as xarray-based data sets, which internally use numpy arrays that encompass the total simulation domain (Fig. 2). Each  $cell(C_i)$  is implemented as a Python object whose xarray-based 145 attributes are references (numpy views) to the corresponding cells of the world object, thereby ensuring memory efficiency and 146 consistency. This structure allows data associated with a particular cell to be accessed and modified either directly through 147 148 the global world arrays (e.g., world.input [1]) or via the respective cell object in the collection world.cells, with both approaches referencing the same underlying data in memory. This ensures that data is always up to date, regardless of the 149 entity, and allows data to be read and modified independently of the hierarchy (world, cell). 150

# 2.3 Bidirectional coupling with LPJmL

To enable a bidirectional, annual data coupling, LPJmL has been extended such that the writing of input and output files can be replaced or extended by TCP/IP socket connections for the corresponding data. This option is now available for all inputs and outputs in the LPJmL configuration. The file lpjml\_config.cjson contains the base configuration of LPJmL, written in a C-flavoured JSON format (Fig. 2). When preprocessed by cpp, the standard C preprocessor, it incorporates additional configurations, such as model inputs, and is parsed to produce a standard JSON file. An excerpt of such a configuration with coupled inputs and outputs is shown as in Fig. A1, also containing additional information such as the name of the coupled model, the start year of coupling, as well as the hostname and the port number of the coupled model. The coupling itself is

172173

174175

180

186

188

190

provided by the LPJmL internal coupler library, which was implemented to encapsulate any socket communication in LPJmL.

At the start of a coupled LPJmL simulation, a connection to the coupled model is established via a connect () call within this library to a host where the coupled model is running with the corresponding port number. The coupling follows a prescribed protocol, as described in Fig. S1 of the Supplement.

Conversely, to enable simple and practical coupling with LPJmL, we have developed pycoupler, a Python interface that provides tools around the coupling as well as handling of LPJmL in Python environments (Fig. 2). It serves a configuration interface for coupled as well as stand-alone LPJmL simulations as well as functionality to send and receive xarray-based data objects from LPJmL. The coupling follows the prescribed coupling protocol, for which pycoupler contains the module LPJmLCoupler. The protocol is provided in detail in Fig. S1 of the Supplement. With LPJmLCoupler, LPJmL is coupled on the Python side, and both sending and receiving methods are provided. It creates the underlying data structures (input, output, grid, country, etc.) and in the case of input data, initialises it with data read in at the start of the simulation. While all static data (e.g., grid, country) is only sent at the start of the simulation, time transient LPJmL output data is received from LPJmL on an annual basis and updated during simulation time via pycoupler (Chapter 2.3). Likewise, the input data can be changed within copan:LPJmL and sent back to LPJmL vice versa. The exchange format LPJmLDataSet is based on the existing data class xarray. DataSet of the Python library xarray (see chapter 2.2) and combines it with principles of LPJmL metadata processing described in Breier et al. (2024). This way, both the received outputs and the inputs that are sent are handled as a single object. The underlying outputs/inputs are in turn available as LPJmLData objects, analogous to xarray.DataArray, exposing each by its LPJmL identifier (e.g. output.soilc or input.landuse). This allows the use of any arithmetic operation and functionality available by xarray and/or numpy. Both libraries are widely adopted and serve many functionalities, such as plotting, statistical analysis, or writing data as Network Common Data Form (NetCDF) or Comma Separated Values (CSV) files. pycoupler also allows for configuring (coupled) LPJmL simulations by providing an LPJmLConfig module, which simplifies reading, handling, modifying and writing of LPJmL configurations, the files of which are subsequently used to run the corresponding LPJmL simulations following Breier et al. (2024). Fig. 3 shows an example of how such a configuration can be set up.

If running an LPJmL simulation locally or submitting it to an High-Performance Computing (HPC) cluster with Simple Linux Utility for Resource Management (SLURM) workload manager support (Yoo et al., 2003), the LPJmLCoupler module is used for data exchange. Sending and receiving data is possible during the runtime of LPJmL on an annual basis. After a defined coupling year, LPJmL waits until the required input data is sent via LPJmLCoupler to continue the simulation until the end of the year to send back the output data. For the following years, the procedure is repeated. For a coupled simulation, the modules are usually applied together and in sequence in an individual run script. This way pycoupler can cover numerous simulation and coupling cases. An example that builds upon the configuration of Fig. 3 is demonstrated in Fig. 4.

The pycoupler package is available on GitHub with more detailed function documentation and examples (Breier and von Bloh, 2025). Beyond that, pycoupler serves further utility functions around LPJmL, such as getting the neighbour cells of a cell or subsetting the grid for country-specific LPJmL simulations.

```
from pycoupler.config import read_config
1
2
                 # Read base configuration from LPJmL
                 config_coupled = read_config(
                     model_path="./LPJmL",
                     file_name="lpjml_config.cjson"
                 # Set coupled run configuration
                 config_coupled.set_coupled(
                     sim_path = ".",
                     sim_name="coupled_run",
                     dependency="historic_run",
                     start_year=2001, end_year=2100,
                     coupled_year=2023,
                     coupled_input=["with_tillage"],
                     coupled_output=["soilc_agr_layer", "harvestc"]
                 # Regrid by country and update configuration
                 config_coupled.regrid(country_code="NLD")
                 # Write a configuration as json file
                 config_coupled_fn = config_coupled.to_json()
```

Figure 3. Configuration of a coupled simulation with LPJmL via the LPJmLConfig module of pycoupler. A base configuration is read in as a LPJmLCoupler object and changed for a coupled simulation, including a regridding to simulate the Netherlands only.

```
from pycoupler.run import run_lpjml
2
                    from pycoupler.coupler import LPJmLCoupler
                    # Run lpjml simulation with socket coupling set
                    run_lpjml(config_file=config_coupled_fn)
                    # Establish coupler connection to LPJmL
                    lpjml = LPJmLCoupler(config_file=config_coupled_fn)
9
                    # Get initial data of previous years of simulation
                    inputs = lpjml.read_input()
                    outputs = lpjml.read_historic_output()
                    for year in lpjml.get_sim_years():
                         # Placeholder to interact with inputs
                        lpjml.send_input(inputs)
                        lpjml.read_output(outputs)
                    lpjml.close()
```

Figure 4. Execution of a coupled simulation with LPJmL via the LPJmLCoupler module of and run function of pycoupler that is based on the coupler extension (Chapter 2.3). inputs and outputs are objects of class LPJmLData and can be accessed and edited following Hoyer and Hamman (2017); Breier and von Bloh (2025).

#### 2.4 copan:LPJmL, a World-Earth modelling framework

The copan:LPJmL framework with its corresponding pycopanlpjml library unites all presented software components pycopan-core (Chapter 2.1), LPJmL (Chapter 2.2) including the coupler library and the Python interface pycoupler (Chapter 2.3)). With that, we propose a novel WEM framework that enables researchers from diverse disciplinary fields to engage in modelling human-environmental interactions in the land-use sector and allows them to specify custom decision rules, agent interactions, or feedback mechanisms, without requiring modification of the LPJmL model. This lowers the entry barrier and fosters interoperability, supporting a wide range of applications and interdisciplinary collaboration.

Within copan:LPJmL, LPJmL, the DGVM, defines the spatial scope and spatial resolution in this setup. It constitutes the ENV taxon of the WEM, and is complemented by the taxa MET and optionally CUL (Fig. 8). While MET processes are typically represented in many integrated modelling approches (i.e., IAMs), the CUL taxon is, as indicated in Chapter 1, often neglected (Schlüter et al., 2017; Beckage et al., 2022). We here provide a platform to include and explicitly represent these social processes of different actors and their interactions that are highly relevant for ENV interactions and feedbacks, such as land use and agricultural management. It facilitates the representation of "the social" from different perspectives. The CUL taxon can cover individual decision-making, social learning, but also economic dynamics, policy-making, and much more. Within the copan:LPJmL framework, MET and CUL processes can be represented with the full flexibility of copan:CORE. Their potential interactions with ENV are bound to the data interface of the terrestrial Earth, the inputs and outputs of LPJmL (Chapter 2.3). These are, on the one hand, the exposed inputs, represented in the World (World.input), and Cell entity (Cell.input). In general, all available time-variable inputs in LPJmL can be exposed via pycoupler and thus coupled, an overview is given in Table 1.

Table 1. Overview and categorization of key inputs to LPJmL (ENV)

| Category      | Key Variables                                                                                     |  |
|---------------|---------------------------------------------------------------------------------------------------|--|
| Climate       | Temperature (daily mean, min, max), precipitation, radiation, atmospheric CO <sub>2</sub> concen- |  |
|               | tration, wind speed                                                                               |  |
| Land Use      | Land systems, irrigation, irrigation systems                                                      |  |
| Management    | Tillage systems, crop residue management, fertilizer & manure nitrogen, sowing dates              |  |
|               | & harvest dates                                                                                   |  |
| Society       | Population density, country affiliation, water use of household, industry & livestock             |  |
| Miscellaneous | Soil texture type, soil acidity, elevation, livestock density, lakes & reservoirs, river          |  |
|               | drainage direction, neighbour irrigation network, water demand for households, live-              |  |
|               | stock and industry, atmospheric nitrogen deposition                                               |  |

A corresponding change to one of these inputs, which is applied in LPJmL in the following year, results in a corresponding biophysical response that is reflected in the outputs. Vice versa, the range of available LPJmL outputs can be utilized in the

218219

- MET and CUL taxon via World (world.output) and Cell (cell.output) to inform and shape the respective processes.
- Here, too, all outputs are available in copan:LPJmL, the most important of which are listed in Table 2.

**Table 2.** Overview and categorization of key LPJmL outputs (ENV)

| Category            | Key Variables                                                                               |
|---------------------|---------------------------------------------------------------------------------------------|
| Vegetation          | Aboveground biomass, total vegetation carbon, net primary productivity, gross primary       |
|                     | productivity, respiration,                                                                  |
| Soil & Litter       | Soil carbon & nitrogen content per soil layer, soil temperature per layer, maximum thaw     |
|                     | depth, N2O emissions, nitrogen leaching, ammonium volatilization                            |
| Water               | Soil evaporation, transpiration, canopy interception, local runoff, river discharge, poten- |
|                     | tial evapotranspiration, soil moisture, irrigation water use per crop                       |
| Fire & Disturbance  | Burned area fraction, fire emissions, fire carbon released, deforestation emissions         |
| Land use & Agricul- | For each simulated crop: yield, growing area, sowing date, harvest date, fertilizer         |
| ture                | amount, fertilizer & application                                                            |

In principle, inputs can be extended, for example, by converting further parameters or settings into inputs. Any state variable or flux can also be written as an output, providing even more opportunities for model coupling. By design, copan:LPJmL allows the integration of a broad range of modelling paradigms of different domains that represent socio-ecological dynamics and feedbacks at varying levels of complexity and abstraction. Table 3 shows an overview of approaches that potentially could be represented in CUL and MET using copan:LPJmL. These include ABMs for simulating heterogeneous actors and interactions, rule-based systems for capturing institutional or behavioural heuristics, optimization-based approaches for identifying efficient or goal-oriented management strategies, and surrogate or machine learning models for data-driven decision-making. System Dynamics models can be incorporated by formulating their underlying equation systems within the Dynamical Systems paradigm, ensuring compatibility with its feedback-oriented architecture. The flexibility enables the exploration of diverse real-world processes ranging from farmer decision-making to governance interventions.

Table 3. Modelling paradigms and application domains supported by copan:LPJmL, with representative examples.

| Paradigm                  | Application Domain(s)                                                                                                                                                                                                                          | Representative Examples                                                                                                                                                                                                                                                                                         |
|---------------------------|------------------------------------------------------------------------------------------------------------------------------------------------------------------------------------------------------------------------------------------------|-----------------------------------------------------------------------------------------------------------------------------------------------------------------------------------------------------------------------------------------------------------------------------------------------------------------|
| Agent-Based Models (ABMs) | Socio-ecological systems such as land-use change and adaptation, or household and (farmer) decision-making; socio-hydrological systems, such as mitigation of extremes and water management; higher-level organisation, e.g., policy scenarios | InSEEDS: heterogeneous farmer agents with LPJmL feedback (Schwarz et al.); CRAFTY: competition-based land allocation and land (Murray-Rust et al., 2014); Household-level farming responses (Wens et al., 2020; Huber et al., 2022); Social-hydrological studies (Schrieks et al., 2021; Kreibich et al., 2025) |
| Rule-Based Models (RBMs)  | Systems that follow rule-based logic, such as agricultural management and land use; higher-level organisations, e.g., acting schemes of governmental institutions                                                                              | Crop Calendar (Minoli et al., 2022a, 2019):<br>Land use allocation (Liang et al., 2021; Verburg<br>and Overmars, 2009); Crop rotation and fallow<br>scheduling (Szalai et al., 2014; Li et al., 2021)                                                                                                           |
| Optimization Models       | Economic land and resource optimization; yield and input efficiency; food security and environmental trade-offs; policy-guided scenario analysis                                                                                               | Examples: MAgPIE, GLOBIOM, or IMAGE (Dietrich et al., 2019; Krey et al., 2020; Stehfest, 2014). E.g. for water and nutrient optimization (Blanco-Gutiérrez et al., 2013; Bodirsky et al., 2012; Beier et al., 2025) or land-based mitigation (Doelman et al., 2020; Bauer et al., 2020)                         |
| Dynamical System Models   | System-level feedbacks and tip-<br>ping dynamics; coupled socio-<br>environmental processes; bioeco-<br>nomic models; evolutionary game<br>theoretic models                                                                                    | Tipping point and regime shift models (Bauch et al., 2016), Socio-epidemic models, complex contagion (Horsevad et al., 2022) Human-climate models (Bury et al., 2019)                                                                                                                                           |
| Data-Driven Models        | Emulators; Pattern discovery; Sce-<br>nario validation and calibration;<br>remote-sensing based approaches                                                                                                                                     | Statistical crop yield emulators (Liu et al., 2023); Surrogate models (Natel et al., 2025); Remote-sensing (Kou-Giesbrecht et al., 2024; Dantas de Paula et al., 2020)                                                                                                                                          |
| LLM-Enhanced Models       | Decision emulation; Stakeholder reasoning; Adaptive behavior                                                                                                                                                                                   | LLM-based farmer management decisions  LLM agents as institutional policy-makers  (Zeng et al., 2025)                                                                                                                                                                                                           |

228229

230231

251252

In addition, the framework supports a wide range of application areas that interact with the land system, in particular its ecological conditions, and resources. This can well result in more complex (social) structures in which, for example, only a first layer of agents interacts with the land system, while other actors are only connected to them, as in a supply and demand model for agricultural products involving farmers, food producers, and consumers. In general, potential domains are land-use change, agricultural management, climate adaptation, food-water-energy dynamics, policy evaluation, or socio-ecological transitions (Table 3). Through the modular interface and configurable data exchange with LPJmL, copan:LPJmL can serve as a backbone for integrated assessments across spatial scales and decision contexts.

#### 3 Application examples

In order to illustrate a broad range of modelling approches that can be covered using copan:LPJmL, we present in the following three different representative applications examples of varying complexity, which are based on the different model paradigms and application areas shown in Table 3. While those examples give a taste of the possibilities of copan:LPJmL, they are not exhaustive, and many more applications of the framework are possible.

#### 3.1 InSEEDS: a new agent-based World-Earth model

In the field of social-ecological system (SES) science, different modelling approaches have emerged to capture the intertwined dynamics between human and biophysical spheres (Farahbakhsh et al., 2022; Anderies et al., 2023; Ye et al., 2024). Arguably, 240 the most prominent approach to social-ecological modelling, ABM, is ideal to be used for modelling with copan:LPJmL 241 242 (Rounsevell et al., 2012; Filatova et al., 2013; Schulze et al., 2017; Donges et al., 2020). ABMs simulate interactions between agents and their environments over time. These agents can represent individuals, households, organizations, or other entities. 243 244 The simulation of these micro-level interactions gives rise to different macro-level outcomes, like spatial adoption patterns of a certain land use (Murray-Rust et al., 2014). Several features of ABMs make them a particularly useful methodological choice 245 246 for the investigation of SES. SES are often understood as complex adaptive systems and therefore are inherently characterized by dynamical adaptation to changing behaviors and environments (Preiser et al., 2018). Furthermore, ABMs facilitate the study 247 248 of macro-level phenomena emerging through micro-level dynamics. Lastly, ABMs are able to capture agent heterogeneity in human and biophysical spheres (Schlüter et al., 2021). InSEEDS, first described in detail in Schwarz et al., is a WEM created 249 250 using the copan:LPJmL framework and uses an ABM component to capture farmer agent decision-making (Fig. 5).

InSEEDS was originally designed to investigate the SES dynamics at play in transitions from conventional farming to regenerative farming practices such as conservation tillage. The CUL taxon comprises farmer management decision-making processes that are based purely on social interactions (i.e., evaluation of social norms) as well as information obtained from social-ecological processes (i.e., observing the environment). The social network of farmers forms the basis for these social dynamics and is initialized based on the LPJmL model grid (Chapter 2.2). In this realization, one representative farmer agent is assigned to each cell. This means that each farmer has a maximum of eight direct neighbours who form their neighbourhood, which is currently the only social network represented. It follows the implementation of acquaintance networks as described

**Figure 5.** Detailed scheme of InSEEDS following Fig. 8 representing the feedback mechanisms among the taxons ENV, MET and CUL. Individuals who are either of type pioneer or of type traditionalist observe crop yields and soil carbon and make decisions on conservation/conventional tillage to then be re-evaluated again.

for copan:CORE (Donges et al., 2020). An example of an implementation of a social-ecological feedback for a farmer in InSEEDS is given in Fig. 6. It shows a simple attitude formation process of farmer agents as an evaluation of their farming performance. The generic processing of any LPJmL output, like agricultural soil carbon or CFT-specific crop yield demonstrates the flexibility copan:LPJmL provides in combination with the multiple features of xarray in subsetting and aggregating the underlying data. Thereby, it allows for direct feedback functions to be set up to simulate important social-ecological aspects, like the attitude of farmers towards their land.

Such processes are part of the MET taxon, constituting the cross-section of socio-cultural and biophysical processes. Fig. 6 shows a typical example of an observation of ecological variables, as well as the initiation of potential management decisions: Information on average crop yield and topsoil carbon content is calculated by the LPJmL ENV component and provided via cell.output to farmer agents within a MET process. Vice versa, farmers' decisions are forwarded to LPJmL as input for the simulation via the MET component. Following the agent-based logic, the current main actors in the InSEEDS model are individual Farmer agents. The Farmer agent class itself is a child class of the Individual agent class in the copan:CORE modelling framework, inheriting the logic of Individuals described in (Donges et al., 2020). As an additional property of the Farmer, we introduce two agent-functional types (AFTs) (Arneth et al., 2014), a traditionalist and a pioneer farmer,

```
1
        def attitude_own_land(self):
2
            self.soilc = self.cell.output.soilc_agr_layer.values[0]
3
            self.cropyield = self.cell.output.harvestc.values.mean()
4
            attitude_soil = (self.soilc - self.soilc_past) / self.soilc_past
            attitude_yield = (self.cropyield - self.cropyield_past) / self.cropyield_past
            return sigmoid(
                self.weight_yield * attitude_yield
10
                 + self.weight_soil * attitude_soil
11
12
13
```

**Figure 6.** Simple example function of a farmer agent estimating their attitude toward their land based on two ENV variables (LPJmL output, see Chapter 2.3). The agent compares the topsoil carbon content and average crop yield between simulation steps, multiplied by the underlying AFT-specific weight. Variables such as soilc\_past and yield\_past refer to values from the past evaluation, while the sigmoid function normalizes the return value (Schwarz et al.).

who differ in their respective weighting of different parameters in the decision-making function (Fig. 6. The decision-making process is based on a formalization of the Theory of Planned Behaviour originally described by Ajzen (1985).

InSEEDS can simulate social-ecological model dynamics on a wide spectrum of scales up to global scale. Analysis possibilities include distributed and accumulated social and ecological outcomes of variables such as attitude, social norm, or soil carbon, crop yield, or even adoption patterns of certain management practices. InSEEDS is the first model realisation using copan:LPJmL that simulates the aforementioned coevolutionary social-ecological dynamics through closed feedback loops. Fig. 10 illustrates the underlying coevolution via three variables: The top row (a) shows the spatio-temporal dynamics of management practice adoption globally. The middle and lower row (b, c) depict the biophysical response of these behaviour changes in topsoil carbon content and average crop yield, compared to a business-as-usual simulation. In some areas, such as Kazakhstan, the spreading of conservation tillage and the underlying coevolution is particularly evident. Here, the adoption started in the southern regions and spread north-eastwards with moderate increases in soil carbon and significant increases in average crop yield. This synergistic effect results in a certain irreversibility in the modelled system. In our simulation, we find that in many regions, the adoption of conservation tillage has a positive effect on these variables, even though the results vary strongly at the local level. To better understand the underlying dynamics of specific cases, a more regional perspective is needed, which can be found together with the decision equations, parameterisation, parameter sensitivity, and detailed simulation results on coevolutionary model dynamics in Schwarz et al..

The development of InSEEDS is currently in an early phase. Future projects such as mapping non-local networks, social systems with multiple layers of complexity, and more social-ecological feedback processes can build on this approach. At this point, at the latest, the connection to the IAMs and the existing paradigmatic problems as described in (Chapter 1), such as

**Figure 7.** InSEEDS simulation of **a**) global spreading of conservation tillage represented as years since last management switch (backwards from simulation end year 2100). The adoption and spatial spreading of conservation tillage is depicted in orange, and conventional farming adoption in blue. If the saturation of both colours decreases over time, this indicates that there will be no further change in strategy. Vice versa, the more saturated the colour, the more recent the change. **b**) and **c**) show the corresponding changes in topsoil carbon content and average crop yield in 2100 compared to a simulation without management changes (business-as-usual) until 2100.

missing closed social-ecological feedback loops can be drawn. With one of these new and rather unique features, it will become apparent whether InSEEDS and the underlying copan:LPJmL approach are capable of generating new insights.

**Figure 8.** Detailed scheme of the crop calendar implementation following Fig. 8 and Minoli et al. (2022a) representing the feedback mechanisms between the taxons ENV and MET. Climate data is forwarded by LPJmL to be used by the Crop Calendar to determine sowing and harvest dates for the upcoming year. Under changing climatic conditions, choosing the right sowing and harvesting times is crucial for maximizing crop yields, indicated by the grey and orange maize plants.

# 3.2 Integrated rule-based approaches: Crop calendar

In many land-use and agricultural systems, management decisions are not made by autonomous agents, but instead follow fixed logics, institutional guidelines, or context-dependent thresholds. These RBMs operate via explicit if—then conditions, fuzzy logic or temporal schedules, allowing for transparent and interpretable decision structures (Arnold et al., 2018; Adriaenssens et al., 2004; Moore et al., 2014). Their structured logic makes them particularly suitable for encoding expert knowledge, empirical heuristics, or scenario-specific governance interventions, especially when interactions between agents are minimal or absent. As such, RBMs offer an efficient and reproducible way to represent adaptive but non-agentive processes across socio-environmental domains. Nevertheless, the boundaries are fluid, and rule-based and agent-based systems often overlap. While global, top-down approaches with Boolean logic can be clearly assigned to RBMs, there are bottom-up, autonomous approaches, such as cellular automata, that can be located between these paradigms (Li et al., 2016). Within copan:LPJmL, the RBM paradigm is particularly useful when system feedbacks should arise directly from dynamic ENV quantities—such as climate or resource availability—rather than from emergent behaviour. This makes them well-suited to simulate adaptive but non-agentic responses to environmental change.

The adjustment of growing seasons in response to changes in climatic conditions is such an example and a central element in agricultural adaptation strategies. While changes in sowing dates may already be implemented by the farmers based on their experience (Waha et al., 2012), cultivar choices are subject to availability and breeding (Zabel et al., 2021). Minoli et al.

```
def calc_warm_winter(self):

self.world.output.temp.mean("time")
min_temp = temp.min(dim="band")

return min_temp > self.basetemp_low

self.basetemp_low
```

**Figure 9.** A simplified vectorized function of the Crop Calendar rule-based model logic, estimating whether a given winter season is considered warm based on multi-year average minimum temperatures.

(2019) proposed a modelling approach for simulating changes in growing seasons based on changes in climatic conditions only. This approach was used to create climate-scenario-specific time series of sowing dates and cultivar parameters as inputs for simulations with LPJmL (Minoli et al., 2022a). Implementing the algorithms of Minoli et al. (2019) using copan:LPJmL allows for a flexible application of adaptive growing seasons during runtime without requiring to previously compute growing seasons and corresponding crop parameters for each climate scenario. In this setup, which is exemplary for rule-based management decisions, LPJmL passes only climate information as output to the growing season rules, which represent the MET taxon. A CUL taxon is not involved in this setup, as there is no interaction between individual rule-based decision-making per crop and grid cell. To compute the required multi-year averages of monthly temperature and precipitation, the data is stored over a 10-year rolling window in world.output or cell.output, updated annually via pycoupler, and continuously averaged during the simulation. Together with a vectorized global xarray-based implementation of the rule set at World level this approach is comparatively concise and computationally efficient compared to its reference (Minoli et al., 2022b). Fig. 9 shows this functionality as a simplified example of the Crop Calendar rule-based model logic.

To verify the suitability of copan:LPJmL as a framework for Crop Calendar, it was implemented in its entirety and applied with one new climate input data set (SSP460, climate model: IPSL-CM6A-LR) from ISIMIP3b (Lange et al., 2024) to reproduce the original approach from Minoli et al. (2022a). Similar to this study, we conducted a comparison of recent and future (2080-2099) sowing and harvest dates for two important crop types, maize and temperate cereals (Fig. 10).

For maize, the implementation dynamically adapts sowing dates in response to climatic changes, enabling earlier sowing in large areas of the temperate latitudes (Fig. 10a), while in the tropical regions, patterns are more heterogeneous, with both earlier and later sowing, depending on local conditions. Maturity dates (Fig. 10b) show greater spatial variability, often, but not always, following the pattern of earlier sowing. In some cases (e.g., India and China), maturity is reached later despite earlier sowing.

For temperate cereals, two varieties –a spring and a winter variety– are distinguished, which are sown in different seasons. The model not only adapts sowing dates for each variety but also allows for variety switching based on climate thresholds. While major variety switches remain rare due to modest warming, the model captures spatially nuanced shifts in sowing timing, with earlier sowing in parts of Canada, Europe, Russia, China, and India, and later sowing in eastern Europe and the

USA. Maturity dates shift accordingly, with earlier maturity in the USA and southern Europe, and later dates, especially in higher latitudes.

These changes emerge from the dynamic rule-based responses to the climate input as indicated in Fig. B1 a and b, showing the change in average annual temperature and annual precipitation between the recent and future time steps. For example, increases in spring temperatures extend the viable growing season and enable earlier sowing in temperature-sensitive regions. The calculation of the maturity date involves a multi-step approach (Minoli et al. (2019) for more details): (i) climate-sensitive harvest rule classification and (ii) harvest date and reason determination based on thresholds such as wet season or the warmest days. Fig. B1 c–f shows that harvest rules for maize shift the harvest reason in response to climate change, for instance, a transition in harvest reason from "mid temperature/precipitation" to "high temperature/precipitation" in South America, and from "mid temperature/mixed" to "high temperature/mixed" in parts of the USA and China (Fig. B1 c and d). Although this is only a comparison between two time steps, the underlying runtime algorithm provides insight into each simulation year.

The results demonstrate that the copan:LPJmL-based crop calendar implementation successfully reproduces the climate-responsive adaptation of sowing and maturity dates of Minoli et al. (2019). However, unlike the reference implementation of Minoli et al. (2019), which requires additional preprocessing steps of the corresponding data products, the copan:LPJmL-based implementation enables a direct, runtime calculation of sowing and harvest dates under changing conditions. This makes the system more suitable for ensemble climate simulations or further SES applications in models such as InSEEDS, where crop production needs to coevolve endogenously to changing biophysical conditions.

Finally, this approach can also be applied to other existing rule-based model systems, such as livestock densities as a function of past grassland performance (Heinke et al., 2023) or whether to plant cover crops in the off-season (Porwollik et al., 2022).

**Figure 10.** Recreated figure after Minoli et al. (2022a) (Fig. 1) showing the difference in simulated sowing **a** and maturing **b** between no adaptation and timely adaptation for a climate period 2080-2099 in an SSP460 scenario using ISIMIP3b data (Lange et al., 2024).

#### 3.3 Enhancing classical modelling with LLMs: LLM fertilization

In recent years, there has been an increasing number of modelling approaches using LLMs to emulate agent behaviour in various use cases, spanning from modelling mobility choices to agents' behaviour in online forums (Gao et al., 2024). This strong synergy with ABM stems from the fact that LLMs are inherently trained to model human language, reasoning, and decision-making patterns (Gao et al., 2024). As such, they are well-suited to serve as proxies for heterogeneous agents, whether individuals, households, or institutions—by generating context-sensitive decisions, goals, or narratives based on inputs from their environment. This capacity makes LLMs particularly compatible with the core idea of ABMs: Simulating the interactions and adaptive behaviour of autonomous entities in a shared environment. However, the integration of LLMs is not limited to ABMs. Their ability to translate between qualitative knowledge and formal rules makes them suitable for enhancing RBMs, for example, by extracting management logic or institutional rules from text sources. Yet, it is in agent-based environments where the conversational and decision-oriented nature of LLMs most directly reflects the modelled processes, making ABMs the current frontier for LLM integration. In the field of land use and agricultural management system, this approach has been taken up by Zeng et al. (2025) to simulate institutional agency of land use dynamics (Chen and Huang, 2024). Using the

**Figure 11.** Detailed scheme of the LLM-based fertilization approach following Fig. 8 representing the feedback mechanisms between the taxons ENV and MET. The LLM gets a prompt with the context of the application, information about location and the cultivated crops, as well as ENV output on crop-specific fertilizer applications and corresponding crop yields. This information helps the LLM to decide whether to increase, maintain, or decrease fertilizer levels for the next simulation year.

copan:LPJmL framework, such an approach can also be used to represent the underlying coevolutionary social-ecological dynamics.

As proof of concept for being able to enhance WEMs with LLM agents, we have developed a simple model for nitrogen fertilizer application based on copan:LPJmL, whose approach is illustrated in Fig. 11. The basis is similar to that of InSEEDS (Chapter 3.1, in which one farming agent is initialized per grid cell to make decisions about local farming practices based on LPJmL (ENV) observations. However, except for the interface between observations and fertilizer application located in the MET taxon, the farmer or decision-making process has been completely outsourced to the LLM. This way CUL processes are not explicitly represented, neither through modelled interactions between farmers, nor through predefined decision rules based on social norms or beliefs. For simplicity, we have therefore decided to omit the CUL taxon in Fig. 11, even though CUL processes might be part of the LLM reasoning. At the beginning of each simulation year, the farming agent is given a prompt like in Fig. 11 (full prompt in Fig. S2) including ENV observations of the farmer and the request to make decisions based on this knowledge. In the fertilization example described here, the initialized LLM-based farming agents know their geographical position, the crops (CFTs) they grow, the share of land they cultivate, the amount of nitrogen fertilizer applied in each of the last 10 years, and the resulting crop yield for the corresponding years. Based on this knowledge, the LLM-farmers decide on the amount of nitrogen fertilizer they want to apply to each of their crops in the next year, to "increase the crop yield by increasing

**Figure 12.** Results of the LLM-based farmer model for Togo in 2050. On the left (a), accumulated nitrogen fertilizer levels between 2024 and 2050 compared to an offline LPJmL run are displayed. On the right, the Nitrogen fertilization (b) and harvest of three crops for one cell (c, marked as red in (a)) are displayed over time.

the application of nitrogen fertilizer as long as it is reasonable" (full prompt in Fig. S2). Moreover, the LLM-farmers are asked to provide their reasoning for the decision taken.

Results of this model for Togo are depicted in Fig. 12. The map (a) shows that the Nitrogen fertilization level varies strongly from cell to cell and thus proves that the implemented LLM-farmers adopt the Nitrogen fertilization level differently depending on their local needs. After the coupling in 2024, the Nitrogen fertilization levels (b) of all three displayed crops are strongly increased by the LLM-farmer with respective higher harvests (c), with the LLM-farmer reasoning "Increased nitrogen for rainfed rice, maize, and tropical cereals due to low historical application and potential yield gains." (full prompt in Fig. S2). After that, the LLM-agents are only making smaller adjustments to the N fertilization levels, reacting to declines in the harvest. While the maize harvest seems to be stable over the simulated time, rice and tropical cereals show more fluctuating harvests. Thus, the LLM Agent in this cell holds the Nitrogen fertilization level stable for maize, while it adjusts its application to the other two crops. This shows that the LLM-farmer can adapt to changes in the conditions without taking unreasonable decisions. While this only constitutes a demonstration case, it illustrates the potential of integrating LLM-based agents with copan:LPJmL to introduce more responsive and context-sensitive management decisions. This may offer a more flexible alternative to static assumptions, such as fixed fertilizer levels—in traditional model configurations.

#### 4 Discussion

A central aim of copan:LPJmL is to provide a platform for connecting modelling paradigms, research domains, and communities (Chapter 3). In particular, we seek to enable the integration of process-rich, dynamic Earth system model components with models of social dynamics, supporting use cases ranging from SES science to IAM applications. Previous studies

detailed Earth system dynamics or the representation of fundamental social systems and their underlying processes (Calvin 400 and Bond-Lamberty, 2018; Krawczyk and Braun, 2025). Conversely, ESMs and DGVMs frequently rely on static or oversim-401 402 plified representations of human behaviour and decision-making (Chapter 1), despite their otherwise detailed process-based structures. The copan:LPJmL framework addresses these gaps by providing a flexible and extensible WEM framework that 403 supports the realization of hybrid approaches, unifying natural and social systems science in a synergistic way (Chapter 3). 404 It allows, for instance, the coupling of agent-based decision-making, rule-based logic, or complex dynamical systems with 405 the biophysical processes of LPJmL. The integrated architecture enables the exploration of social-ecological feedbacks and 406 coevolutionary mechanisms across multiple spatial and temporal scales. We demonstrate this potential with three examples 407 (Chapter 3.1; Schwarz et al., Chapter 3.2, Chapter 3.3). Yet, there are many more opportunities, as the framework offers a 408 high degree of interoperability, making it relatively easy to combine different approaches. Also, combinations are thinkable, 409 e.g., where InSEEDS farmer agents would access crop calendar rules and use this information to make decisions about which 410 crops and varieties to grow. The possibilities within the MET and CUL taxon are manifold and are mainly constrained by the 411 available computing resources. The copan:LPJmL framework builds upon copan:CORE and its modular and open structure 412 413 that has now been enhanced by the LPJmL ENV integration — all without additionally coupling LPJmL within the model code itself. This lowers the entry barrier and fosters interoperability for modellers from diverse fields enabling researchers to 414 incorporate customised decision rules, agent interactions, additional entities such as governance agents or cooperations, and 415 feedback mechanisms, without requiring modification of the LPJmL model. This is also the main advantage compared to many 416 pre-existing coupler libraries (e.g., Hanke et al., 2016; Hutton et al., 2020; Müller et al., 2024). Both the LPJmL coupler li-417 brary and the Python interface pycoupler (Chapter 2.3) were purpose-built to couple LPJmL with the copan:CORE framework, 418 which is the basis of the integrated copan:LPJmL framework and supports the implementation of numerous projects and ideas, 419 420 some of which are illustrated in Chapter 3. This approach eliminates the need to integrate libraries that may require advanced technical know-how —such as BMI (Hutton et al., 2020)— for model coupling. At the same time, copan:LPJmL follows 421 the FAIR principles for research software in providing a findable, accessible, interoperable, and reusable modelling software 422 (Barker et al., 2022), with extensive documentation and tutorials available at copanlpiml.pik-potsdam.de. 423 Nevertheless, there are some obstacles and shortcomings associated with the use of copan:LPJmL. Currently, the coupling 424 between ENV and MET/CUL is only possible on an annual basis, constrained by the LPJmL coupler library and pycoupler 425 (2.3). As with all dynamically coupled modelling frameworks, the development of any of the coupled components requires 426 testing and vetting of the coupled system and eventually co-development. Simulated decision making in MET/CUL does not 427 428 only depend on plausible decision-making mechanisms, but also on plausible and quantitative accurate simulated responses in ENV and vice versa. This complexity often leads to using legacy model versions in coupled systems (e.g., Müller et al., 2016). 429 430 Required testing could be facilitated by a model validation tool chain to allow the harmonization and integration of data sets from various sources such as FAOSTAT (FAO, 2025) for comparing these reference data with simulated data from the coupled 431 system and stand-alone components (e.g. LPJmL) and at a later stage for calibrating the parametrizations of model components. 432 433 As for now, the functionality of copan:LPJmL and each copan:LPJmL-based model presented here (Chapter 3) is backed up

have highlighted key limitations in many agent-based and optimization models, which often lack either the integration of

448

462

465

by unit and integrity tests to verify their functional and internal validity. With copan:LPJmL, we intend to provide a modelling framework to address research questions around the complex dynamics of the Anthropocene and its coevolution of human and natural systems. While we have tested different types of models based on this novel framework (Chapter 3), we expect that more features, revisions and extensions will be necessary for future models built on this framework. The open-source basis of all model components should facilitate necessary changes (Schaphoff et al., 2025; Breier and von Bloh, 2025; Breier et al.,

# 440 5 Summary and Outlook

2025b; Breier, 2025).

This paper introduces copan:LPJmL, a new modelling framework designed to build World-Earth models with a process-rich Dynamic Global Vegetation Model (LPJmL) and a flexible and modular core providing the structures and functionalities to represent various kinds of socio-cultural and socio-metabolic structures and processes. This way, models built on this framework can represent social and biophysical dynamics in a consistent and co-evolutionary manner. With minimal overhead and a lightweight Python interface, copan:LPJmL enables the coupling of LPJmL to diverse types of decision logics—from topdown rule-based models to agent-based bottom-up dynamics, without the need for modifying the LPJmL model itself. By integrating LPJmL as the single component of the ENV taxon into the copan:CORE framework and hierarchy, the framework supports easy access to LPJmL outputs and enables dynamic adjustments to inputs via flexible coupling mechanisms. Future work could build on this setup and extend the current ENV taxon by additionally representing atmosphere and ocean dynamics, for example, by using the Potsdam Earth Model (POEM, Drüke et al., 2021), allowing it to develop the framework towards a more comprehensive human-Earth system model of the Anthropocene. The three examples presented and discussed in this paper illustrate the breadth of modelling approaches that copan:LPJmL can accommodate: from the top-down, vectorized Crop Calendar, to the bottom-up, agent-based InSEEDS, and towards novel, experimental LLM-based setups. These diverse implementations underscore the framework's adaptability to a wide range of research questions related to land-use dynamics, adaptation, and the co-evolution of human and natural systems. By adhering to FAIR principles and providing extensive documentation, copan:LPJmL invites collaboration across research domains and communities. It lowers technical barriers for incorporating social dynamics into process-based Earth system modelling and creates a space for new perspectives on topics such as food security, land-use resilience, and regenerative transformations. Ultimately, the framework aims to support the growing need for integrated tools that enable better understanding—and shaping—of coupled human–Earth dynamics in the Anthropocene.

Code availability. The copan:LPJmL framework is composed of four software components, each maintained in its own GitHub repository, and the version applied in this paper is archived on Zenodo. LPJmL is available at https://github.com/pik-lpjml/LPJmL(Schaphoff et al., 2025), pycoupler at https://github.com/pik-lpjml/pycoupler (Breier and von Bloh, 2025), pycopancore at https://github.com/pik-copan/pycopancore (Breier et al., 2025b), and pycopanlpjml at https://github.com/pik-copan/pycopanlpjml (Breier et al., 2025c). All components are licensed under the GNU General Public License v3.0, except pycopancore, which is distributed under the BSD 2-Clause License. The

- models InSEEDS as well as Crop Calendar and LLM Fertilization (both part of landmanager library), have been developed under the same license and are available at https://github.com/pik-copan/inseeds (Breier et al., 2025d) and https://github.com/jnnsbrr/landmanager (Breier et al., 2025a). Comprehensive documentation of copan:LPJmL, including installation instructions, tutorials, a complete API overview, and usage examples, is available at https://copanlpjml.pik-potsdam.de (Breier, 2025).
- Data availability. The historical climate data set (GSWP-W5E5) that has been used for the InSEEDS and LLM fertilizer simulations as well as the future scenario data set, used for the Crop Calendar simulations (SSP460, IPSL-CM6A-LR) are both available on the ISIMIP homepage https://data.isimip.org/1 (Lange et al., 2023, 2024). All further data is either linked directly to the model that is archived together with the model code, model outputs, and scripts that have been used to produce the results presented in this paper on Zenodo (https://doi.org/10.5281/zenodo.17054847, Breier and Prawitz, 2025).

# 475 Appendix A: Configuration of LPJmL

```
"coupled_model": "landmanager",
2
                      "coupled_host": "localhost",
                      "coupled_port": 2224,
                      "start_coupling": 2025,
                      "input": {
                           "with_tillage": {
                               "id": 7,
                               "socket": true
10
11
                      },
                      "outputs": {
                               "id": "harvestc",
                               "file": {
                                    "socket": true
```

**Figure A1.** Excerpt of the LPJmL base configuration file lpjml\_config.cjson with coupled model name (coupled\_model), host (coupled\_host), and port (coupled\_port), start year of coupling (start\_coupling) as well as input and outputs to be coupled.

# 476 Appendix B: Additional figures

**Figure B1.** Additional crop calendar variables: (a) Global temperature and (b) precipitation change from 2025 to 2090 in the SSP460 scenario using ISIMIP3b (Lange et al., 2023). Harvest rules (c, d, e, f) following Minoli et al. (2019) for both CFTs, temperate cereals and maize for 2025 and 2090. Harvest reason (g, h, i, j) also follows Minoli et al. (2019) with a similar order.

Author contributions. JB conceived and designed the study. JFD, DG, and CM contributed to the conceptual design of the framework and supervised the work. JB, WvB, and HP led the development of the copan:LPJmL software as well as the models and application examples. SBW, MB, and LS contributed to model and software development. JB, SBW, and HP performed model simulations, analysed data, and visualised results. JB and LS led the writing of the manuscript. CM, SBW, HP, DG, WvB, and JFD contributed to writing and manuscript revision. All authors discussed the simulation results and reviewed and edited the final manuscript.

Competing interests. At least one of the (co-)authors is a member of the editorial board of Geoscientific model development.

Acknowledgements. J.B., L.S., H.P., D.G., and J.F.D. gratefully acknowledge financial support from the Generation Foundation, the Global 484 Challenges Foundation, and Partners for a New Economy via the Earth4All project, as well as from the European Union's Horizon 2.5 -Climate Energy and Mobility programme under grant agreement No 101081661 (project WorldTrans), J.F.D. and J.B. acknowledge support 485 from the project "CZS Research Groups for Earth System Modelling" funded by the Carl-Zeiss-Stiftung, J.F.D. acknowledges funding from 486 487 the European Research Council Advanced Grant project ERA (Earth Resilience in the Anthropocene, ERC-2016-ADG-743080) and from the project CHANGES funded by the German Federal Ministry for Education and Research (BMBF) under grant 01LS2001A. The authors thank 488 489 the European Regional Development Fund (ERDF), the BMBF, and the Land Brandenburg for providing resources on the high-performance computing system at the Potsdam Institute for Climate Impact Research. The authors used LLM tools, such as ChatGPT, to improve the 490 readability of the manuscript; all content was subsequently reviewed, edited, and approved by the authors, who take full responsibility for 491 the published work. 492

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
