# Peer review of "copan:LPJmL: A new hybrid modelling framework for dynamic land use and agricultural management"

_EGUsphere, 2025_

## Author Comment (AC1)

**Review #1**

This manuscript introduces a newly developed modelling framework copan:LPJmL, which can simulate a wide range of socio-ecological dynamics. This work makes an important contribution to addressing the conceptual and technical challenges in modelling intertwined human and nature processes. The overall structure of the manuscript is clearly presented. The three examples introduced well highlight this framework's flexibility and versatility.

Thank you very much for the extensive positive and constructive feedback. Below we provide a point-by-point response with suggested changes to improve the manuscript accordingly.

However, the manuscript could be improved by distributing the content more properly between the manuscript and the technical manual to enhance the ease of understanding for readers. In general, I would suggest leaving the conceptual content, modelling decisions/justifications, and model logic in the manuscript, while the technical specifications in the GitHub/Zenodo repo, so that readers can easily navigate from high-level conceptual description to implementation details. More suggestions follow here.

We agree on a better differentiation between conceptualization and API documentation.  The API documentation clearly belongs to the software documentation/technical documentation in the software repositories (and hosted on the website copanlpjml.pik-potsdam.de) itself. In the revision, we will carefully evaluate all described entities, remove overly technical ones and retain only those essential for  explaining the model structure in the paper (e.g. World, Cell,  Input, Output).

1. copan:LPJmL is defined as a framework in the manuscript. It might be ambiguous to readers as a framework can mean different things, such as a conceptual framework, a model, a sub-model, or a wrapper for the existing LPJmL to interface with copan:CORE. It would be best to clarify what copan:LPJmL exactly is early in the text. This could also help readers know clearly what kind of contribution the manuscript is intended to stress.

Thank you, the term "framework" indeed requires clarification. We will explicitly define copan:LPJmL early in the Introduction, highlighting that it extends the copan:CORE modelling framework by integrating the biophysical Earth system model (LPJmL) via a coupling interface as the ENV taxon. This clarification will specify its role as neither a stand-alone model nor a conceptual framework, but as a software and modelling framework enabling tightly coupled human–Earth system simulations with process-based and spatially-explicit detail.

2. Usually, readers are motivated to invest efforts in looking into technical details (e.g., code) only if the concepts are attractive and clearly presented. A class diagram appears unable to serve as an efficient means of communication, as it normally provides neither conceptual simplicity for broad readers nor accurate technical details for developers. It would be best to replace the class diagram with a conceptual framework that illustrates the connections and information flow between model components.

The class diagram will be moved to the appendix and replaced in the main text by a conceptual diagram illustrating model components, data flow, and feedbacks between LPJmL, copan:CORE entities, and above all the user-defined model

3. Accordingly, much of the technical description, especially the parts that mix narratives with variable names defined in the code, could be organised into a README.md/txt file in the code repo, leaving only the description of the conceptual idea and model logic (using text, pseudocode, or equations) in the paper. I saw the current code repo only has a minimal technical description in contrast with that in the manuscript. A proper redistribution of the content might be necessary.

We agree and will redistribute content accordingly. Code-specific descriptions, API-level details, and variable-level explanations will be moved to the software documentation (Read the Docs and GitHub repositories). The manuscript will be revised to focus on conceptual design, modelling logic, and coupling principles. The online documentation will be expanded to ensure that all technical details previously described in the manuscript remain accessible.

4. The code shown in the figures seems to provide limited information about the model mechanism. If the code only serves as examples of the use of copan:LPJmL, the README file in the repo should be the best place.

The code snippets shown in the manuscript are intended solely as illustrative examples demonstrating the flexible interface and versatility that enables different modelling approaches. We see that this perspective is limited to Python developers and modellers and therefore will be removed or replaced by a conceptual figure. Detailed model mechanisms belong to the respective model implementations (e.g. InSEEDS) and are documented in their dedicated repositories, which we will cite explicitly.

5. The manuscript could be improved by avoiding relying on Python syntax in the narrative. Although Python and object-oriented programming are widely used in research, using syntax-dependent expressions like "world.input[1]" might not be an ideal way to express the gist of copan:LPJmL.

We agree that Python-specific syntax in the narrative may obscure the conceptual structure of the coupling framework. We will therefore revise the manuscript to remove implementation-dependent expressions (e.g. world.input[...]) from the main text. The underlying functionality will then be described using language-agnostic pseudocode and a schematic flow diagram that emphasize the abstract world-state concept and the annual exchange of inputs and outputs between ENV and MET/SOC.

6. It is fantastic to learn that copan:LPJmL can accommodate a range of modelling approaches. However, the three examples do not contain enough information about how this framework can achieve this. Developers or model users might not be clear about what efforts they should make to switch modelling approaches within copan:LPJmL. I would suggest describing the examples using an identical structure: for each example, including background information, concepts, model processes, model settings, and outcomes, while highlighting the role of copan:LPJmL and what model users should do. In addition, please cite the source code of each example in the text.

Thank you for this helpful suggestion. We will restructure all example sections following a consistent template including: (i) background and motivation, (ii) Setup and & configuration, and (iii) Illustrative outcomes. We will also cite the corresponding source code repositories in the text.

7. Line 287, the citation "Schwarz et al." is not complete.

This is a paper under review that is closely linked to this one as it builds on copan:LPJmL and describes the InSEEDS model in detail. We will correct the citation, i.e. its current preprint version.

8. Line 370, ")" is missed in "(Chapter 3.1".  In addition, I am not sure whether "section" or "chapter" is more accurate in the context. Please check the convention regarding the word choice of GMD.

Chapter will be replaced by section and the closing parenthesis will be added.

9. Please cite the sources of the figures and code if they are already publicly available elsewhere.

We will add explicit citations for all figures and code where it is reasonable apart from the code and data availability statement.

---

## Author Comment (AC2)

**Review #2**

This article describes the coupling of the copan:CORE World-Earth model and the DGVM LPJmL, giving three examples of its use for agent-based, rule-based and LLM-based modelling. It's great to see this approach being developed further and the resulting copan:LPJmL framework has real promise for modelling the interactions of human and natural (land) systems, and doing so in ways that are distinct from established models. The easy availability of the framework is also important, and it should be useful to many.

While I think the article is strong, there are several points that either could or need to be strengthened in a revision. These include conceptual descriptions of some aspects of the approach, but also how some of the core functionalities are operationalized. I've highlighted these in the comments below.

We appreciate the reviewer's careful reading of the manuscript and the detailed and thoughtful comments provided. The feedback has been very helpful in identifying where conceptual clarity, interpretation, and applicability of the copan:LPJmL framework can be strengthened. Below, we respond point by point to the reviewer's comments and outline how we plan to revise the manuscript accordingly.

General comments

- The novel capabilities of the model are given in overview, but the conceptual details aren't always apparent. There's an emphasis on technical description (largely appropriate given the journal) but this isn't always convincingly linked back to the objectives. In particular more content on how (and which) social and decision processes can be modelled would be really useful. There are inevitably some parts that are unclear at the moment, but also I think some under-selling in that the authors leave more open questions about applicability than they need to. See specific comments below on these points.

We agree with this assessment and will strengthen the conceptual framing in the revised manuscript, in particular by more explicitly linking the technical components of copan:LPJmL (entities, interfaces, and coupling mechanisms) to the classes of social and decision-making processes they are able to represent. We will expand the discussion of the three example applications to clarify how different decision paradigms are already illustrated: InSEEDS implements individual-level decision-making based on the Theory of Planned Behaviour (norms, attitudes, perceived control); the Crop Calendar represents rule-based management decisions; and the LLM fertilization example demonstrates an alternative form of decision emulation. Beyond these specific cases, we will more clearly articulate that the framework is not restricted to farmer-level decision-making, but can represent a wide range of behavioural theories (from homo oeconomicus over bounded rationality to more

complex theories like the Value-Belief-Norm Theory or the Social Identity Approach), as well as higher-level social processes such as policy-making, governance, market dynamics, and supply chains. We will further explain how these theories could be represented by copan:LPJmL. This will allow us to better highlight both the current capabilities and the broader applicability of copan:LPJmL for modelling social and institutional dynamics in coupled human–Earth systems, and to reduce the degree to which the manuscript leaves its potential under-specified.

- Resolution: The default resolution given here is fairly coarse for some of the processes (especially social) that the authors highlight as being important; things like social norms and networks, but also diverse vegetation growth within what are represented as single land holdings. Some more detail and justification of the implications of resolution, and how they can be accounted for in an applied model, is needed.

This is an important and well-taken observation. In the current implementation, the 0.5° grid and one agent per cell should be understood as representative of multiple real-world actors within a cell summarizing their decision-making . This design choice was made primarily to demonstrate the breadth of modelling approaches that copan:LPJmL can accommodate, rather than to provide a direct representation of land-holding structures or fine-scale social networks. In the revised manuscript, we will add a clearer discussion of the implications of spatial resolution for representing social norms, networks, and heterogeneous management, and of how this can be addressed in applied studies. In particular, copan:LPJmL allows placing multiple agents within a single grid cell. This will clarify how users can refine the spatial and social resolution depending on their research questions.

- Figures: I'm not convinced how useful the figures comprising code (or components, as in Fig. 2) are. The code is already available, so it might be best to use more generic/descriptive representations here to give conceptual overviews and to provide more interpretation of model capabilities.

Figure 2 will be replaced by a conceptual framework illustrating model components, data flow, and feedbacks between LPJmL, copan:CORE entities, and the user-defined model. Further figures containing code (see below) will either be replaced by conceptual figures or be removed

- The discussion would also benefit from substantially more reflection on applicability and interpretation. Some of this could perhaps deal with flexibility in the model to represent more things or to represent them differently. Even the authors seem unconvinced at times about this!

We agree and will revise the Discussion accordingly. In particular, we will expand the reflection on what kinds of research questions copan:LPJmL is well suited to address, how flexibility in model design can be used constructively, and how

limitations should be interpreted in relation to modelling goals rather than as deficits. The balance between opportunities and constraints will be made more explicit.

Lines 21-22: 'Committee, 1986' – incomplete reference?

We will complete this reference in the revised manuscript.

Lines 28-29: It might be worth stating how you see WEM relating to social-ecological systems modelling

We will add a short clarification explaining how World-Earth Models relate to, and extend, social-ecological systems modelling by explicitly focusing on coevolutionary feedbacks between human and Earth system processes at larger scales.

Lines 83-84: Repeated 'detailed'

We will revise the wording to avoid repetition.

Line 88: the reference 'Schwarz et al.' doesn't have a year

We will add the missing year in the revised manuscript.

Line 109: It's a bit unclear what 'the framework' is.

The term "framework" requires clarification. We will explicitly define copan:LPJmL early in the Introduction as an extension of the copan:CORE modelling framework by integrating the biophysical Earth system model (LPJmL) via a coupling interface as the ENV taxon. This clarification will specify its role as neither a standalone model nor a conceptual framework, but as a software and modelling framework enabling tightly coupled human–Earth system simulations.

Line 110: Clarify the 'aforementioned taxonomy' by writing out the terms again?

We will restate the relevant taxonomy terms explicitly to avoid ambiguity.

Line 113: Is the word 'entities' being used in the same sense throughout the description? It seems imprecise in its meaning.

We agree and will clarify the use of the term "entity", distinguishing between conceptual entities (e.g., actors, institutions) and their technical realization within copan:CORE.

Lines 115-116: Is the Group entity distinct from the social institutions mentioned previously?

We will clarify the distinction and relationship between Groups as technical entities and social institutions as potential social constructs represented by using the technical "group" entries.

Lines 126-127: It would be useful to have here a description or link to one, of what the various agricultural management practices can be.

This is stated in Table 1, we will point to it here.

Lines 129-130: Some text already at this point on how this resolution aligns with the objectives of the model would be helpful.

We will add text explaining how the chosen resolution supports the intended modelling objectives and what implications this has for interpretation

Figure 2: I'm particularly unsure how helpful this Figure is. There are lots of terms in there that aren't (and can't easily be) interpreted, so add little value. This might be better in SI with full explanations, and a simpler version here with normal language explanations.

Answered above. We will replace the figure by a more conceptual one.

Line 164: 'Serves as a' or 'provides a' alternatively.

We will revise the wording accordingly.

Line 170: 'are only sent...'

We will revise the sentence for clarity.

Lines 195-199: It would be useful to have less technical (more conceptual) summaries above this point to establish that these things can be modelled meaningfully, beyond the technical capacity of the model. But even at that technical level, it's not clear to me how decisions and interactions can/should be modelled at this resolution.

We agree and will add a higher-level conceptual summary before the technical description, clarifying how decision-making and interactions can be meaningfully represented despite the chosen resolution.

Lines 205-206: Are land use and agricultural management given as examples of social processes, or ENV interactions & feedbacks? The following list about the CUL taxon does I think need to be justified and explained better.

Thank you for pointing out this ambiguity. In the revised manuscript, we will clarify the distinction between land use and agricultural management as MET taxon processes, rather than purely social (CUL) or biophysical (ENV) processes. The taxonomy underlying copan:LPJmL distinguishes between ENV processes (purely biophysical dynamics), CUL processes (purely social structures and processes), and MET processes, which form the interface between the two. In this framework, the CUL taxon captures human structures and processes across multiple levels of social organization, ranging from individual decision-making and social learning to institutional, economic, and policy processes. Land use and agricultural management are classified as MET processes because they mediate between CUL and ENV: decisions originating in the CUL taxon (e.g., farmer or institutional choices) are translated into changes applied to the ENV taxon, while ENV responses (e.g., yields or soil conditions) are similarly mediated back to CUL via MET processes such as observation or harvest. We will revise the text to make this conceptual separation and the role of the MET taxon more explicit.

Table 1: The 'Key variables' for Land Use seem questionable – what is included in 'land systems'?

We will revise the table and accompanying text to clarify what is meant by "land systems" and which variables are included.

Table 2: Should the vegetation row include PFTs?

Thanks for pointing this out. While we intended to list general output classes, we do agree that adding information at what structural level these can be provided and processed is helpful here. We will add text to clarify that vegetation is represented by multiple PFTs and CFTs and that soils are represented as distinct layers.

Lines 256-257: I think it would be very worthwhile to describe how this can be extended.

The neighbourhood can either be extended or complemented by non-local networks on national or world level networks, depending on the context where Individuals are communicating. We will clarify this in the main text.

Figure 5: legend – why 'following' Fig 8? (or, why in this order). Also are crop yields and soil carbon the only things observed by the farmer types, and are they known perfectly?

This statement is redundant and will be removed. And yes, crop yields and soil carbon are the only two things observed and known perfectly by farmer agents in the inSEEDS model which is why this sentence will be adjusted to be referred to as field measurements. Measuring inaccuracies of farmers are not considered in this approach.

Lines 266-267: As in previous comment, it seems unnecessarily unrealistic for farmer agents to know these things perfectly in the modelled world, and not to know about other constraints or outcomes. Explain/extend

For this first proof-of-concept version of InSEEDS we chose crop yield and soil carbon as two meaningful system variables for farmers, one for short-term and one for long-term performance. In contrast to crop yield, which farmers can measure accurately, access to soil carbon can be considered a field measurement. We will provide this information in the text. We agree that it is not realistic to assume precise field measurements to be accessible for all farmers. For the purpose of the modelling of the underlying social-ecological feedbacks, we think that it did not make sense to introduce "measurement inaccuracies."

Line 273: Can the decision-making process be varied? Explain/extend

The decision making process is only a formalization in form of code which can be replaced by any theory. We will explicitly state how alternative decision-making processes can be implemented.

Lines 288-289: Anything you can say about how they can do this would be really interesting here.

We will expand this section to include more information on how mapping non-local networks, social systems with multiple layers of complexity, and more social-ecological feedback processes can be realised by using the current structure.

Lines 291-292: This is a very underwhelming end to the example! Surely you can – and need to – argue that they can at least generate some sort of new insight?

We agree and will revise the concluding paragraph of the example to better articulate what kinds of scientific insights such models can generate, such as on coevolutionary dynamics of adaptive management dynamics under climate change, new more tightly linked social-ecological feedbacks in the DGVM, ESM and IAM context.

Figure 8 legend: This description seems quite distinct from real-world decisions, and more of a forward-looking optimisation. I'm not sure how much or easily this can be varied to represent alternative ways of making decisions, but it seems important that it can be.

It does not necessarily need to be decision making process, it's a rule-based approach that has been reimplemented and endogenized on the base of copan:LPJmL. The purpose is to demonstrate the framework's versatility; it can be used to substitute input data generation that requires scenario specific data generation (Minoli et al. 2019, 2022) otherwise.

We see the point, that this figure is very technical and does not show the value of the simple interface copan:LPJmL provides for these kind of studies. We will therefore remove it or replace it by a conceptual figure to visualize the easy access of each world's variable and the computation of corresponding statistics.

Line 381-382: It's good that the reasoning is provided as this can change the decision. Is this checked by a human user?

Yes, we will clarify the role of human oversight and how LLM outputs are interpreted within the modelling context. However, given that thise is a simple proof-of-concept implementation, we did not systematically analyse the sensitivity of the LLM-Model in this application.

Line 424 on: I would prefer more reflection than a description followed by 'shortcomings'. The model of course can't do everything, so the framing here seems overly-negative, but then interpretation of what the model actually can do is largely missing.

In line with the above suggestion, we agree and will revise the discussion to place limitations in a more balanced context, focusing on scope of applicability and interpretation rather than shortcomings alone.

Summary and Outlook: This is quite repetitive of the first part of the discussion; I think you could save space here and cover more ground overall, especially guiding the reader in what the technical capabilities of the model can be used for.

We will revise this section to reduce repetition and focus more strongly on guiding readers toward potential applications and future research directions enabled by the framework.